# Freeze-Omni: A Smart and Low Latency Speech-to-speech Dialogue Model with Frozen LLM

Xiong Wang [* 1]  Yangze Li [* 1 2]  Chaoyou Fu [3]  Yike Zhang [1]  Yunhang Shen [1]  Lei Xie [2]  Ke Li [1]  Xing Sun [1]
Long Ma [1]

## Abstract

The GPT-4o's excellent duplex speech interaction ability has given users an impressive experience. Researchers have recently proposed several multimodal LLMs to achieve user-agent speech-to-speech conversations. In this paper, we propose a novel speech-text multimodal LLM architecture called Freeze-Omni, and our main contribution is that the speech input and output modalities can be easily connected to a textual LLM while keeping the LLM's parameters frozen throughout the training process. We effectively ensure that the intelligence of the Freeze-Omni in the speech modality is at the same level as that in the text modality of its backbone LLM while achieving low latency in the end-to-end spoken response. In addition, we also designed a method to achieve duplex dialogue ability through multitask training, giving Freeze-Omni a more natural style of dialogue ability between users and agents. In summary, Freeze-Omni holds great potential to conduct speech-to-speech dialogue based on a multimodal LLM under the condition of a frozen LLM, avoiding the catastrophic forgetting problem caused by limited data and training resources.

## 1. Introduction

Recent years have witnessed a rapid development of large language models (LLMs). The LLM family represented by the GPT series (Floridi & Chiriatti, 2020; Achiam et al., 2023) of OpenAI has demonstrated extraordinary capabilities. As speech interaction is one of the most natural forms of human-computer interaction, combining speech input and output with an LLM can bring a natural experience to

users. The traditional method adopts a cascaded approach of ASR + LLM + TTS to achieve interaction with LLM in the speech modality. However, this engineering-centered pipeline approach often leads to considerable interaction latency. Nevertheless, GPT-4o (OpenAI, 2024) has changed this situation – it provides an end-to-end speech interaction mode which has significantly improved user experience, triggering a research boom regarding multimodal LLMs for speech-to-speech interaction.

In the field of general LLMs, many public models such as Llama 3.2 (Dubey et al., 2024), Qwen2.5 (Team, 2024), Mixtral (Jiang et al., 2024), etc., have provided good opportunities for researchers to develop downstream tasks. Therefore, in the field of multimodal LLMs targeting speech-to-speech conversation, works such as Mini-Omni2 (Xie & Wu, 2024b), LLaMA-Omni (Fang et al., 2024), Moshi (Défossez et al., 2024) and GLM-4-Voice (Zeng et al., 2024) have provided excellent references for researchers. These works adopt different strategies to align the speech modality with an LLM and design some tricks to achieve a duplex dialogue mode.

In this research context, we found that in the process of aligning the LLM with the speech modality in existing public speech-text multimodal LLMs (Chu et al., 2024; Défossez et al., 2024; Fang et al., 2024; Fu et al., 2024; Zhang et al., 2023; Xie & Wu, 2024a; Zeng et al., 2024), the parameters of the LLM are more or less fine-tuned. However, in most cases, it is challenging for researchers to easily collect spoken Q&A data at the million-hour level (the corresponding text content can be comparable to the amount of data for training text-modal LLM). Fine-tuning the LLM inevitably brings the catastrophic forgetting problem to the LLM, negatively impacting its original "intelligence" ability. In addition, only a few works have evaluated the performance of spoken question-answering tasks for speech-to-speech multimodal LLMs, showing an obvious gap in the performance between spoken Q&A and text-modality Q&A. Therefore, in this paper, we propose a speech-to-speech dialogue LLM called Freeze-Omni, which achieves effective speech-text modality alignment while keeping the LLM parameters frozen, obtaining low-latency speech dialogue

---
[*]Equal contribution  [1]Tencent, China [2]ASLP@NPU, China
[3]Nanjing University, China. Correspondence to: Long Ma <malonema@tencent.com>.

*Proceedings of the 42nd International Conference on Machine Learning*, Vancouver, Canada. PMLR 267, 2025. Copyright 2025 by the author(s).

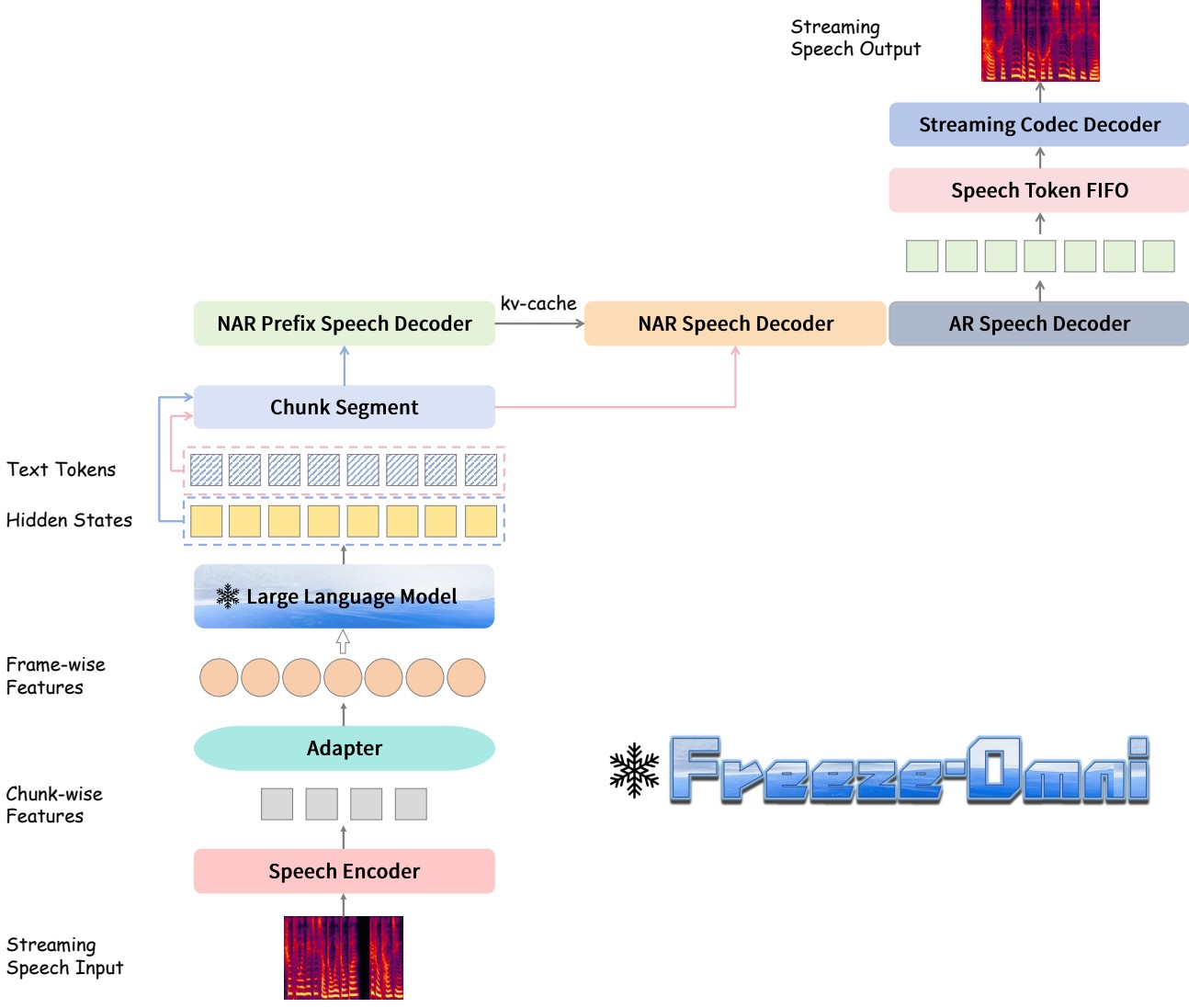

*Figure 1.* Overview of proposed Freeze-Omni. The streaming speech input forms chunk-wise features through the speech encoder, and then is connected to the LLM through the adapter. The LLM generates hidden states and text tokens, which are sent to the NAR prefix speech decoder and the NAR speech decoder in the form of chunks, respectively, after chunk segmentation. Finally, the AR speech decoder sends the generated tokens into the speech token FIFO, and the streaming codec decoder generates streaming speech output from the FIFO according to a fixed speech token chunk size.

capabilities while maintaining the original intelligence of the backbone LLM. Specifically, Freeze-Omni is mainly implemented in the following steps:

**Modeling of speech input**    We first use a large amount of ASR data to align the speech encoder and the LLM, enabling the LLM to understand the semantic information from the speech. Then, with the LLM frozen, a training strategy of prompt embedding is used to let the model have the ability to possess speech input to text output, training on only a small amount of Q&A data.

**Modeling of speech output**    Second, we use a sizable amount of text-speech paired data to train the AR-based speech decoder, which can generate speech tokens from text, and a single-codebook based codec model is used to decode the speech token into a waveform. Then, we design a prefix kv-cache fine-tune strategy, using the hidden state vector output by the LLM to transfer the speech decoder into the output text space of LLM, achieving the ability of

text input to speech output while keeping the LLM frozen.

**Design for duplex dialogue** Finally, we connect the speech encoder and speech decoder from the above parts to the backbone LLM. Then, a task of chunk-wise state prediction is used to determine whether or not the user interrupts the dialogue, achieving the duplex speech-to-speech dialogue ability.

In conclusion, the main contributions of Freeze-Omni are as follows:

1) The parameters of the LLM are completely frozen throughout the training process of Freeze-Omni, maintaining the original intelligence of the LLM and achieving low-latency speech-to-speech dialogue at the same time.

2) The paired text-speech Q&A training data is at a small scale and consumes fewer computing resources in the building of Freeze-Omni.

3) Freeze-Omni can support any (multimodal) LLM with a text modality and retains the abilities of the LLM, such as prompt following and role-playing. Moreover, if it is necessary to change the style of the LLM's response, it is only required to fine-tune it with text data in the corresponding style.

## 2. Model

### 2.1. Overview

Freeze-Omni is a speech-to-speech dialogue model, and its architecture is shown in Fig. 1, exhibiting the characteristic of being "smart" as it is constructed upon a "frozen" text-modality LLM. This enables it to keep the original intelligence of the LLM backbone, without being affected by the forgetting problem induced by the fine-tuning process for integration of the speech modality. Specifically, Freeze-Omni contains a speech encoder that supports streaming speech input and a speech decoder that generates streaming output speech. During the training process, Freeze-Omni first achieves alignment between speech input and text output, then between text input and speech output. Finally, by connecting these two components to the LLM, the ability of speech input to speech output is obtained. This section will provide a detailed introduction to the architecture, training strategy, and duplex dialogue design of Freeze-Omni.

### 2.2. Modeling of speech input

#### 2.2.1. CHUNK-WISE STREAMING SPEECH ENCODER

To allow Freeze-Omni to support speech input and achieve a rapid and low-latency response to input speech, it utilizes a chunk-wise streaming speech encoder to transform the input speech features into a high-dimensional representation. Then, an adapter module maps the high-dimensional

representation into the embedding space of the backbone LLM. The speech encoder module here consists of several down-sampling convolution layers and several Transformer (Vaswani et al., 2017) blocks, while the adapter only comprises several down-sampling convolution layers. The reason for using down-sampling is to reduce the frame rate of the speech features, increase the speed of the LLM in the prefill stage, and decrease the latency.

#### 2.2.2. TRAINING STRATEGY

A 3-stage training strategy shown in Fig. 2 is used for the speech encoder, allowing Freeze-Omni to acquire the ability to understand the streaming input speech while keeping the LLM frozen.

1) The first stage shown in Fig. 2(a) is the same as the training process of a common speech recognition model. The input is speech features, and the label is the transcript corresponding to the speech. CTC (Graves et al., 2006) is used as the loss function.

2) In the second stage shown in Fig. 2(b), we use the speech encoder trained in the first stage as the initialization parameter and connect it with the LLM utilizing an adapter. The output of the LLM still uses the transcript corresponding to the input speech as the label. Several trainable special tokens are added to the input part to guide the LLM in completing the training process at this stage. In this stage, except for the frozen LLM, the parameters of other networks are all trainable.

3) In the last stage shown in Fig. 2(c), we first construct a dataset of multi-round questions and use the LLM backbone relied on in the training to generate multi-round answers. The dataset constructed in this way will be completely compatible with the LLM backbone. Subsequently, we use a multi-speaker TTS system to generate data in the speech modality for the questions part and add trainable prompt embedding before each question in the multi-round to guide the LLM to achieve the ability of speech input to text output. In this stage, the trainable special tokens in stage 2 will be dropped, only the prompt embedding part is trainable and they use the same parameters for each question, the speech encoder is frozen to maintain the acoustic robustness obtained from stage 2, and the LLM is also frozen to ensure that its intelligence is not affected.

### 2.3. Modeling of speech output

#### 2.3.1. ARCHITECTURE

Inspired by VALL-E (Chen et al., 2024), Freeze-Omni uses a token-based speech decoder that contains NAR prefill and AR generation stage to achieve speech output capabilities. The speech decoder mainly consists of the NAR decoder, the AR decoder, and the decoder of a codec model. Both the

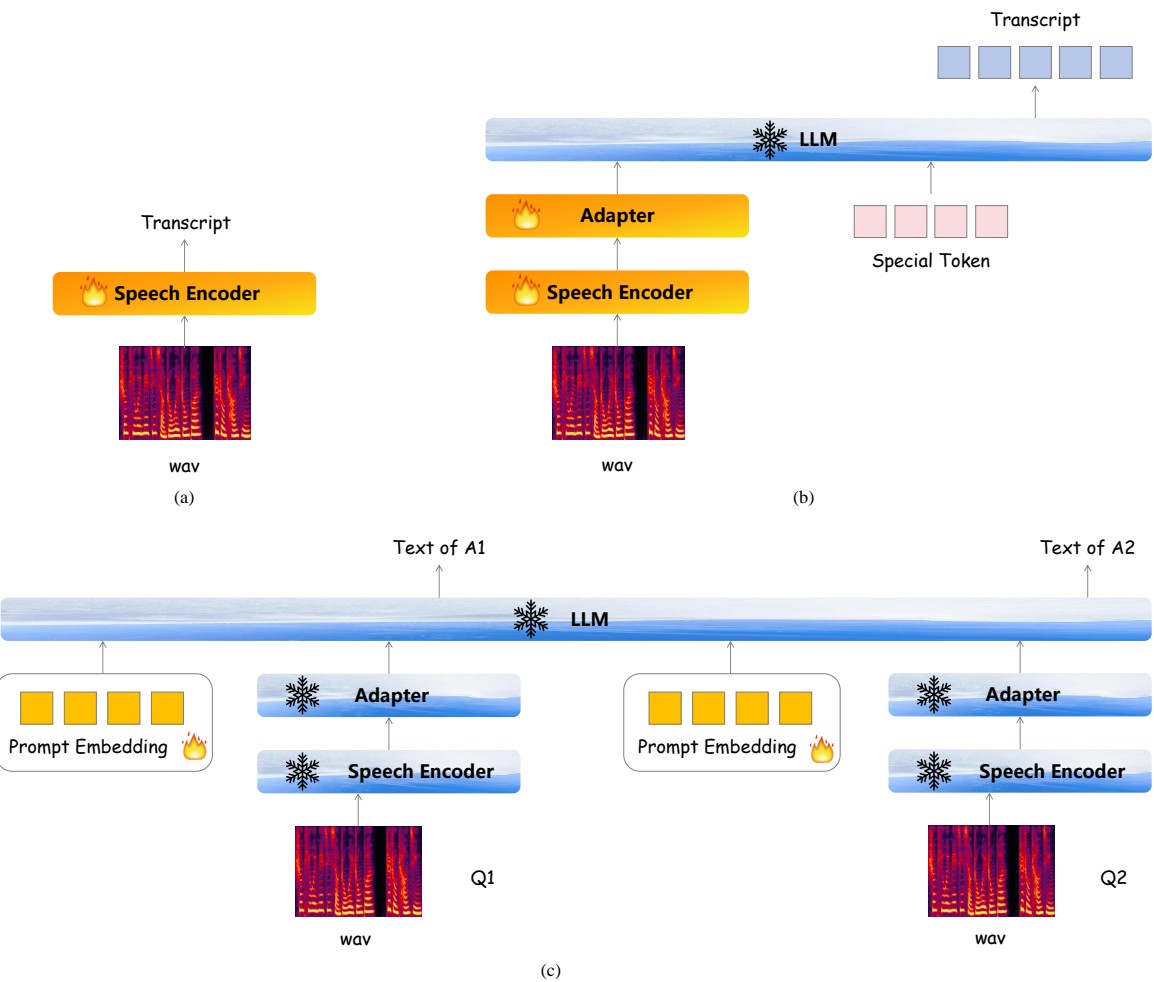

*Figure 2.* The 3-stage training method for modeling of speech input, the speech encoder in (c) is used in Freeze-Omni finally and LLM is frozen in all 3-stage.

NAR decoder and AR decoder are built upon transformer blocks. The NAR decoder is used to model the semantic features from the output of LLM, and then the AR decoder generates speech tokens based on the output of the NAR decoder. Finally, a decoder of the codec model converts the speech tokens into a speech stream.

### 2.3.2. TRAINING STRATEGY

For the modeling of speech output, we still use a 3-stage training method as shown in Fig. 3, enabling Freeze-Omni to obtain the ability of generate speech from the output of LLM while keeping the LLM frozen.

1) As shown in Fig. 3(a), we first train a single-codebook-based codec model using only speech data. Since a single codebook is sufficient for extracting speech tokens from the speech signal of a limited number of speakers, using a single codebook here can reduce the complexity and latency of the

system as much as possible.

2) In the second stage shown in Fig. 3(b), we first construct a large amount of text-speech paired data and pass the text through the tokenizer of the backbone LLM to convert the text into text tokens. Then, we pass the text tokens through the embedding layer of the LLM to convert them into embedding vectors as semantic features and send them to the NAR speech decoder. The AR speech decoder predicts the output speech tokens in the form of teacher force. The labels here are extracted using the codec model trained in stage 1. The NAR and AR speech decoders use the same parameters, and the embedding layer of the LLM is frozen.

3) In the last stage, we use the same multi-round questions and answers data set in stage 3 of Sec. 2.2.2 and use the text tokens and hidden state sequence generated by the backbone LLM. As shown in Fig. 3(c), an additional NAR prefix speech decoder is added to model the hidden state of the

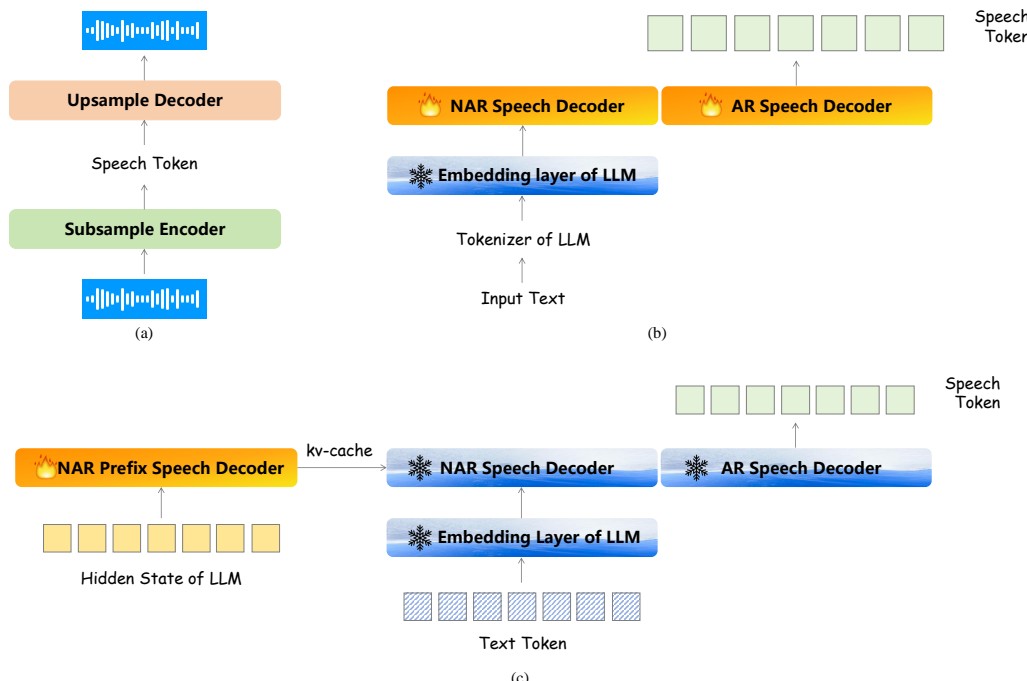

*Figure 3.* The 3-stage training method for modeling of speech output, the speech decoder in (c) is finally used in Freeze-Omni, and LLM is frozen in all three stages.

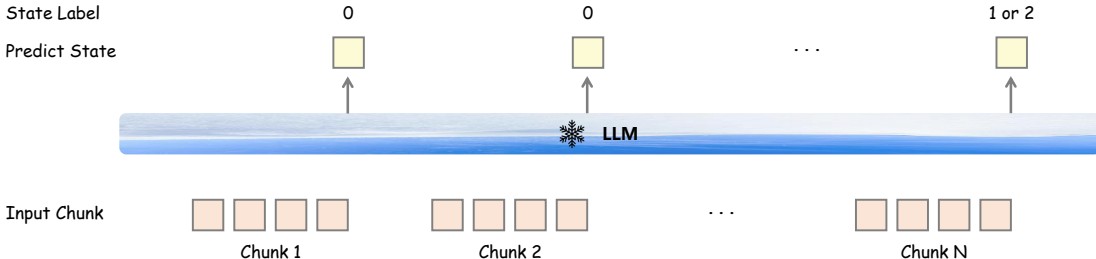

*Figure 4.* Method of chunk-level state prediction used in the prefill stage of the LLM. An additional classification layer is added to the output hidden state of the LLM corresponding to the last frame of each chunk output by the speech encoder to predict the state.

LLM and pass its output kv-cache to the NAR speech decoder. Then the text token will be fed to the NAR speech decoder trained in stage 2. The text token label for the AR speech decoder is the speech data produced by the output text of the LLM using a TTS system and converted into speech tokens by the codec model in stage 1. In this stage, the NAR prefix speech decoder uses different parameters from the NAR and AR speech decoders, and only the parameters of the NAR prefix speech decoder are trainable, while the parameters of other networks are frozen. Because the style of the text tokens produced by the LLM is different from that of the text in the large amount of text-speech paired data obtainable in stage 2, the significance of the third stage lies in closely coupling the speech decoder with the output of the LLM to reduce the occurrence of bad cases.

### 2.4. Design for duplex dialogue

After the above training process, Freeze-Omni has the ability to convert speech input to speech output. However, to better approximate the natural form of speech-to-speech dialogue, we use multi-task for chunk-level state prediction as shown in Fig 4. We first use an acoustic VAD[1] module to detect the starting point of the streaming speech. When the VAD is triggered, the speech stream will be sent into Freeze-Omni chunk by chunk, and an additional classification layer will be added after the last layer of the LLM to predict different states. Three states are defined here: state 0 indicates that the current LLM can continue to receive speech, and states 1 and 2 indicate that the current chunk is the end of the speech.

---

[1]https://github.com/snakers4/silero-vad

State 1 means that the user will interrupt the dialogue and the LLM will perform a new generation stage; State 2 means that there is no need to interrupt the conversation. Both states will stop sending speech streams to Freeze-Omni and reset the VAD module. The training process of this part is completed in stage 3 of Sec. 2.2.2, using a multi-task method to optimize the cross-entropy loss of both the state classification layer and the LLM. It should be noted that the state labels here are only valid on the last frame of each chunk.

In addition, we used a "model as a server" strategy to implement the speech-to-speech dialogue system. First, we started several models simultaneously and regarded them as a server. Then, when a user's VAD was triggered, the speech would be sent to the server in the form of chunks, and the server would be responsible for scheduling which idle model should respond to the current chunk. Since we separated all the kv-cache and CNN cache of the speech encoder and LLM during the inference process, the server only needs to save the inference cache for each user. In this way, any model on the server could respond to any chunk of any user, and there was no need to specify which model was used as a monitor or a generator.

# 3. Experiments

## 3.1. Setups

### 3.1.1. DATASETS

In this paper, we only randomly selected 60,000 multi-round Q&A data from *moss-003-sft-data* [2] and used the backbone LLM to generate new answers to replace its original one. We used a zero-shot TTS system to synthesize its text into speech. For the modeling of speech input of Freeze-Omni, we used 110,000h of internal speech-text paired ASR data, including both Chinese and English, in stage 1 and stage 2. In stage 3, we used the pairing of speech input and text output of the multi-round Q&A data mentioned above. For the modeling of the speech output of Freeze-Omni, we used about 3,000h of text-speech paired data generated by a zero-shot TTS system in stages 1 and 2. In stage 3, we used the pairing of text input and speech output of the multi-round Q&A data mentioned above.

### 3.1.2. MODEL CONFIGURATION

*LLM backend*    For experiments in this paper, we used Qwen2-7B-Instruct[3] as our backbone LLM. As an outstanding 7B-level public LLM, it is beneficial for us to verify

our method. Besides, Freeze-Omni can use any LLM as a backbone because its training process does not update any of the LLM's parameters.

*Speech Encoder*    We used a multi-layer convolution with 4-times downsampling and 24 layers of transformers with a hidden size of 1024. The adapter consists of a multi-convolution layer with 2 times downsampling. The number of parameters for the speech encoder is approximately 350M, with an output frame rate of 12.5 Hz. The input of the speech encoder is the mel-filter bank feature with a 25ms window size and 10ms shift.

*Speech Decoder*    We used TiCodec[4] (Ren et al., 2023) as the codec model, and we customized the configuration so that the size of the codebook is 1024 with a single-codebook and the frequency of the speech token 40Hz. For the speech decoder part, the NAR (Prefix) speech decoder and the AR speech decoder are 4-layer Llama decoder layers with a hidden size of 896. The number of parameters for the speech decoder is approximately 120M, and the output sample rate of the codec model is 24000Hz.

### 3.1.3. TRAINING

In the training process, we used the Adamw (Loshchilov & Hutter, 2017) optimizer with a warm-up learning rate scheduler, and different learning rates were used in different stages. The learning rates used in the three stages of the modeling of speech input are 2e-4, 1e-4, and 6e-4 respectively. The learning rates used in stages 2 and 3 of the modeling of speech output are both 5e-5, and the training hyperparameters used in stage 1 are the same as those in TiCodec. All the experiments were completed on 8 GPUs.

## 3.2. Results on speech input

To measure the understanding ability of Freeze-Omni for input speech, as shown in Tab. 1, we verified the accuracy of ASR on different evaluation sets for the model in stage 2 of the modeling of speech input. Since the parameters of the speech encoder and adapter used in stage 3 are unchanged compared to those in stage 2, it can be considered that these results can represent the input speech understanding ability of Freeze-Omni. In the training of stage 2, we used a dynamic chunk training method (Yao et al., 2021) to enhance the robustness of the model so that different chunk sizes can be used in stage 3. From the results, it can be seen that in the case of dynamic chunk training, decoding with $chunk = \infty$ shows better performance compared to $chunk = 4$. If dynamic chunk training is not used but $chunk = 4$ decoding is used, better results can be obtained, but this also means that the chunk size cannot be changed in stage 3. In this paper, to pursue the best performance, all experiments are

---

[2] https://huggingface.co/datasets/fnlp/moss-003-sft-data
[3] https://huggingface.co/Qwen/Qwen2-7B-Instruct

[4] https://github.com/y-ren16/TiCodec

*Table 1.* The ASR performance of the model corresponding to stage 2 in the modeling of speech input, where {aishell-1 (Bu et al., 2017),test_net (Zhang et al., 2022), test_meeting (Zhang et al., 2022)} are Mandarin evaluation sets, measured in CER (%), while {dev-clean,dev-other,test-clean,test-other} (Panayotov et al., 2015) are English evaluation sets, measured in WER (%).

| Model | aishell-1 | test_net | test_meeting | dev-clean | dev-other | test-clean | test-other |
|---|---|---|---|---|---|---|---|
| Wav2vec2-base (Baevski et al., 2020) | - | - | - | 6.0 | 13.4 | - | - |
| Mini-Omni2 (Xie & Wu, 2024b) | - | - | - | 4.8 | 9.8 | 4.7 | 9.4 |
| VITA-1.5 (Fu et al., 2025) | 2.16 | **8.4** | **10.0** | 3.3 | **7.2** | 3.4 | **7.5** |
| **Freeze-Omni** | | | | | | | |
| + $chunk = \infty$ | **2.15** | 8.57 | 10.09 | **3.29** | 7.4 | **3.24** | 7.68 |
| + $chunk = 4$ | 2.79 | 12.6 | 14.2 | 4.16 | 10.21 | 4.05 | 10.48 |
| + w/o dynamic | 2.48 | 11.8 | 13.46 | 4.03 | 9.45 | 3.82 | 9.79 |

*Table 2.* The CER(%) of the speech decoder on 1,000 evaluation utterances under different $top$-$k$.

| Method | $top$-$k$ | | | | |
|---|---|---|---|---|---|
| | 1 | 2 | 3 | 4 | 5 |
| Speech Decoder w/o Prefix | 5.27 | 4.64 | 4.76 | 4.66 | 5.03 |
| + pre-network | 3.11 | 2.75 | 2.77 | 2.84 | 2.94 |
| Speech Decoder | 3.9 | 3.65 | 3.53 | 3.62 | 3.71 |
| + pre-network | 2.19 | 1.69 | 1.85 | 1.9 | 1.99 |

completed on the model with this configuration of the last row in Tab. 1.

### 3.3. Results on speech output

Because we investigated the speech-out performance of Freeze-Omni in a single-speaker case in this paper, we randomly selected 1,000 utterances of text tokens and hidden states output by the LLM as the input of the speech decoder and compared the ASR accuracy of the synthesized speech with the label text. As shown in Tab. 2, the performance of the model in stage 2 of the modeling of speech output (Speech Decoder w/o Prefix) and the model in stage 3 (Speech Decoder) under different AR decoding parameters $top$-$k$ are presented respectively, and CER (%) is evaluated using *paraformer-zh*[5] (Gao et al., 2022). From the results, it can be concluded that after introducing the hidden state of the LLM as the input of the NAR prefix speech decoder, the speech decoder can be more completely aligned with the LLM, reducing the occurrence of bad cases and get a lower CER (%). In addition, the increasing $top$-$k$ shows better robustness of the speech decoder with a prefix fine-tune because a larger top-k means a higher quality requirement on the posterior probability distribution output by the model.

In addition, the NAR and AR speech decoders need to model the LLM embedding outputs and speech tokens simultaneously, but the spaces represented by these two are different.

Therefore, to verify whether the generation quality would be improved if the NAR speech decoder had additional parameters for modeling the outputs of the LLM embedding layer compared to the AR speech decoder, we added an extra pre-network between the NAR speech decoder and the LLM embedding layer. This pre-network consists of two Llama decoder layers with the same configuration as the NAR speech decoder. As shown in Tab 2, this method can significantly improve the speech quality generated by the speech decoder.

### 3.4. Results on spoken question answering

To demonstrate the intelligence of Freeze-Omni, we verified the accuracy of spoken question answering on three sets: LlaMA-Questions[6] (Nachmani et al., 2023), Web Questions[7] (Berant et al., 2013), and Trivia QA[8] (Joshi et al., 2017). Since Web Questions and Trivia QA only have text, we used the *edge-tts*[9] tool with voice at *en-US-BrianNeural* to synthesize them into spoken modality. Tab. 3 shows the accuracy of Freeze Omni and its used backbone LLM Qwen2-7B-Instruct on these three sets. From the results, it can be observed that Freeze-Omni exhibits excellent performance compared to other models because the accuracy gap between it and the backbone LLM is smaller than that of

---

[5] https://huggingface.co/funasr/paraformer-zh

[6] https://github.com/google-research-datasets/LLAMA1-Test-Set

[7] https://huggingface.co/datasets/Stanford/web_questions

[8] https://nlp.cs.washington.edu/triviaqa/

[9] https://github.com/rany2/edge-tts

*Table 3.* The accuracy (%) of different models in question answering on three sets. The models in the first four rows all use speech as input, while the models in the last two rows use text as input. The backbone LLM of Freeze-Omni is Qwen2-7B-Instruct, and the backbone LLM of Moshi is Helium. Both Freeze-Omni and Qwen2-7B-Instruct use greedy search in the generation stage with zero-shot, and the accuracy is calculated using the output text. Except for Freeze-Omni and Qwen2-7B-Instruct, previous evaluation results are derived from corresponding references.

| Model | Modality | Web Q. | LlaMA Q. | Audio Trivia QA |
|---|---|---|---|---|
| SpeechGPT (7B) (Zhang et al., 2023) | Audio&Text | 6.5 | 21.6 | 14.8 |
| Spectron (1B) (Nachmani et al., 2023) | Audio&Text | 6.1 | 22.9 | - |
| Moshi (7B) (Défossez et al., 2024) | Audio&Text | 26.6 | 62.3 | 22.8 |
| GLM-4-Voice (9B) (Zeng et al., 2024) | Audio&Text | 32.2 | 64.7 | 39.1 |
| **Freeze-Omni (7B)** | Audio&Text | **44.73** | **72** | **53.88** |
| Helium (Défossez et al., 2024) | Text Only | 32.3 | 75 | 56.4 |
| Qwen2-7B-Instruct | Text Only | 45.13 | 77.67 | 63.93 |

*Table 4.* Detailed information of statistical latency. Among them, 50% represents the median, and 90% represents the percentile at 90. The unit of the results in the table is (ms). All results are completed using pytorch with bfloat16 inference.

| Latency description | Avg. | 50% | 90% |
|---|---|---|---|
| LLM interrupted → LLM generate first text token chunk | 478 | 468 | 750 |
| First text token chunk → Prefill of speech decoder | 15 | 15 | 17 |
| Prefill of speech decoder → Generate first speech token chunk | 237 | 235 | 252 |
| First speech token Chunk → Decode first PCM chunk | 11 | 11 | 13 |
| Total | 745 | 753 | 1020 |

Moshi, which also verifies that Freeze-Omni has the same level of intelligence in text and speech modalities. It is worth mentioning that thanks to the freezing of LLM during the training process, Freeze-Omni's performance on the spoken question answering task even surpasses that of GLM-4-Voice, which uses speech training data far more than Freeze-Omni.

### 3.5. Analysis on end-to-end latency

To verify the latency of Freeze-Omni for speech-to-speech dialogue, we defined two parts of latency, namely statistical latency and non-statistical latency. The statistical latency refers to the time from the LLM being interrupted to the first PCM chunk of speech generated. Specifically, it can be divided into four parts as shown in Fig 4, these results are based on a speech token chunk size of 40 and the use of text token chunk segmentation based on the sentence-split strategy. The non-statistical latency refers to the time from the real endpoint of speech to the LLM outputting the interrupt state. This part needs to be measured manually and cannot be counted automatically. According to our case analysis conclusion, the non-statistical latency is about one to two speech encoder chunk sizes, and according to the experiment configuration above, this time is about 160ms to 320ms. In summary, if we consider the influence of

network latency (about 200 to 300ms), the average latency of Freeze-Omni used in real scenarios will be controlled at about 1.2 seconds. This result means Freeze-Omni can deliver a low-latency speech-to-speech dialogue experience for users.

## 4. Conclusion and future work

In this paper, we propose Freeze-Omni, a text-audio multi-modal LLM capable of low-latency speech-to-speech dialogue, which does not need fine-tuning the LLM backbone, showing excellent performance in various tasks, especially in the spoken question answering. In the future, to explore more speech dialogue capabilities, we plan to do the following:

- We will upgrade the speech encoder to a general audio encoder to complete tasks like emotion understanding and audio captioning.
- Under the condition of a frozen LLM, we will add more tasks to make the LLM complete more downstream tasks in speech dialogue, like the state prediction ability.
- We plan to support multiple voices and instruction-following ability in the speech decoder part so that it can obtain more instruction information from the hidden state of the LLM and provide richer speaking styles.

## Acknowledgements

Supported by National Natural Science Foundation of China under Grant No. 62441234. Supported by the AI & AI for Science Project of Nanjing University under Grant No. 2024300529.

## Impact Statement

This paper presents work whose goal is to advance the field of Machine Learning. There are many potential societal consequences of our work, none which we feel must be specifically highlighted here.

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
