# OpenReview forum: "Freeze-Omni: A Smart and Low Latency Speech-to-speech Dialogue Model with Frozen LLM"
_ICML.cc/2025/Conference — ICML 2025 poster_

### Official Review · Reviewer_RJrj · 2025-03-07

**Overall Recommendation:** 2

**Summary:**

The paper proposed Freeze-omni, which can enable speech input and output capabilities for any LLM backbone without tuning its parameters. In such a case, the model can achieve end-to-end chat experience without losing the intelligence behind the LLM backbone. The framework consists of multiple speech encoder and decoder modules to be aligned with LLM, training with different stages, e.g. speech input modules for understanding, speech output modules for generation, and chunk-level state prediction for interruption. The highlight of the performance is the high accuracy on several speech synthesized language benchmark compared to other speech LLM models, which benefits from the freezing LLM.

**Claims And Evidence:**

The core idea of the framework is to maintain the intelligence of LLM by freezing its parameters. The results of high accuracy on language benchmark provided the evidence for such claims.

A potential issue is the claim of omni for the proposed framework. It seems the LLM is expanded with speech modality only. How about the vision modality? The omni is over-claimed considering the model capabilities. I have also noticed that there are many references in this paper using omni but with audio modality only. This can be misleading to the entire community.

**Essential References Not Discussed:**

There are several papers discussing how to enable speech functions for LLM with its parameters freezing. However, these papers do not expand the model with speech output, but I think it is worthing noting these works:

Wang, Chen, et al. "Blsp: Bootstrapping language-speech pre-training via behavior alignment of continuation writing." arXiv preprint arXiv:2309.00916 (2023).

Fathullah, Yassir, et al. "Audiochatllama: Towards general-purpose speech abilities for llms." arXiv preprint arXiv:2311.06753 (2023).

Lu, Ke-Han, et al. "Developing Instruction-Following Speech Language Model Without Speech Instruction-Tuning Data." arXiv preprint arXiv:2409.20007 (2024).

Kang, Wonjune, et al. "Frozen Large Language Models Can Perceive Paralinguistic Aspects of Speech." arXiv preprint arXiv:2410.01162 (2024).

Fan, Ruchao, et al. "AlignFormer: Modality Matching Can Achieve Better Zero-shot Instruction-Following Speech-LLM." arXiv preprint arXiv:2412.01145 (2024).

**Experimental Designs Or Analyses:**

The paper lacks some important comparisons, especially when this paper is more based on empirical study.

1. No comparison to cascades systems. The proposed system is very similar to train an ASR model as LLM input and a TTS model with LLM output. The comparison would provide the community insight and better evident the effectiveness of the method.

2. In terms of the ASR performance in Table 1, pls compare with the model using continuous features as input, e.g. Qwen2-audio, since Freeze-omni is using the similar.

3. It seems that Table 3 presents the speech-to-text QA performance. How about speech in and speech out QA? One of the contributions of the paper compared to previous work is the speech output. The paper should discuss about speech-to-speech QA performance.

4. What is the performance of speech-to-text QA after aligning the speech input module? Is it the same as what is shown in Table 3 after aligning the speech out modules? The question is regarding to whether adding speech output modules can affect the performance for speech-to-text QA.

**Methods And Evaluation Criteria:**

The proposed method is intuitive, and the evaluation criteria is pretty standard in speech/audio LLM research.

**Other Comments Or Suggestions:**

Overall, the paper is more like a technical report and does not provide enough ablation studies to provide insights to the community.

The proposed method is without strong theoretical claims. More experimental study is very necessary to strengthen the paper.

The proposed method involves many stages of training. People might be very interested in how each part affect the performance.

**Other Strengths And Weaknesses:**

Strengths: This work is a great study to expand LLM with speech functions without destroying the language capabilities.

Weakness:

1. I don't like the omni as the model does not include the vision modality. Shouldn't it be Freeze-audio or something?
2. The paper lacks comparison and ablation studies.
3. Reference missing.

**Questions For Authors:**

I have read the paper very carefully and do not have further question.

The authors, please correct my comments if you feel I don't understand it in the right place.

**Relation To Broader Scientific Literature:**

Freeze LLM for enabling speech functions is explored previously in literature. See "Essential References Not Discussed". This paper expands the core idea with speech output.

**Theoretical Claims:**

This is not a theoretical paper, but more based on empirical study.

There is a very fundamental issue that the paper didn't discuss, that is, how would the framework compare to the ASR+LLM+TTS cascades system? There is no such comparison in the experimental section.

---

### Official Review · Reviewer_BFAM · 2025-03-13

**Overall Recommendation:** 4

**Summary:**

The paper introduces Freeze-Omni, a novel speech-text multimodal large language model (LLM) designed for speech-to-speech interaction while keeping the backbone LLM’s parameters frozen throughout the training process. This architecture enables low-latency, end-to-end spoken response while preserving the intelligence of the original LLM, addressing key challenges such as catastrophic forgetting and high computational costs associated with fine-tuning.

**Claims And Evidence:**

Yes. The claims are all supported.

**Essential References Not Discussed:**

None

**Experimental Designs Or Analyses:**

More Comprehensive Evaluation on Speech Input and Output：The paper mainly evaluates the accuracy of speech recognition (ASR) for speech input and the character error rate (CER) for speech output. However, additional metrics such as Word Error Rate (WER) for output speech and intelligibility scores (e.g., MOS – Mean Opinion Score) could provide a more comprehensive assessment of speech quality.

Comparative Analysis with More Baselines：While the paper compares Freeze-Omni with a few existing models, more state-of-the-art speech-to-speech systems (e.g., recent versions of GPT-based multimodal models) could be included for a broader comparison.

Latency Optimization：The paper provides latency analysis but does not discuss potential optimizations. Exploring methods to reduce the response time further, such as improving the efficiency of the speech encoder or speech decoder, could be beneficial.

**Methods And Evaluation Criteria:**

Yes. The methods and evaluation make sense for the problem.

**Other Comments Or Suggestions:**

None.

**Other Strengths And Weaknesses:**

-Strengths:
1. Innovative Frozen-LLM Architecture: The proposed Freeze-Omni model maintains the parameters of the backbone LLM completely frozen throughout training. This prevents catastrophic forgetting and ensures that the model retains the intelligence of the original LLM while integrating speech modalities.

2. Three-Stage Training Strategy: The proposed three-stage training process efficiently models both speech input (ASR) and speech output (TTS) while keeping the LLM frozen.

3. Low-Latency Speech Interaction: The model is optimized for real-time speech-to-speech dialogue, achieving an average response time of 1.2 seconds in real-world scenarios. This is significantly lower than traditional ASR + LLM + TTS pipelines.

4. High Accuracy in Spoken Question Answering: The model achieves competitive performance in spoken Q&A tasks, with an accuracy gap between Freeze-Omni and its text-only backbone LLM smaller than other speech-enabled LLMs like Moshi.

Freeze-Omni presents a well-designed, efficient, and practical approach to integrating speech-to-speech interaction into LLMs while preserving intelligence, reducing computational overhead, and maintaining low latency. The combination of freezing the LLM, modular training strategies, and real-time duplex dialogue capabilities makes it a notable advancement in the development of multimodal conversational AI systems.

-Weaknesses:
1. More Comprehensive Evaluation on Speech Input and Output：The paper mainly evaluates the accuracy of speech recognition (ASR) for speech input and the character error rate (CER) for speech output. However, additional metrics such as Word Error Rate (WER) for output speech and intelligibility scores (e.g., MOS – Mean Opinion Score) could provide a more comprehensive assessment of speech quality.

2. Comparative Analysis with More Baselines：While the paper compares Freeze-Omni with a few existing models, more state-of-the-art speech-to-speech systems (e.g., recent versions of GPT-based multimodal models) could be included for a broader comparison.

3. Latency Optimization：The paper provides latency analysis but does not discuss potential optimizations. Exploring methods to reduce the response time further, such as improving the efficiency of the speech encoder or speech decoder, could be beneficial.

4. Scalability and Adaptation to Different LLMs Not Fully Explored: While the paper claims Freeze-Omni can work with any LLM, it is only tested on Qwen2-7B-Instruct. There is no empirical evaluation of how the approach generalizes to larger models or smaller, more efficient models that might be deployed on edge devices.

5. Training Efficiency vs. Larger-Scale Training: The paper highlights the efficiency of training on 8 GPUs with only 60K Q&A data, but it does not explore whether performance would scale with larger datasets.

**Questions For Authors:**

None.

**Relation To Broader Scientific Literature:**

None

**Theoretical Claims:**

None

---

### Official Review · Reviewer_WoBP · 2025-03-13

**Overall Recommendation:** 1

**Summary:**

This paper proposes a framework to provide a frozen text LLM with spoken dialogue abilities, by integrating it with a speech encoder and a speech generation system. When orchestrated by an auxiliary turn prediction module, this allows for the model to interact along multi-turn conversations with a latency lower than one second. While easily turning text LLMs into spoken dialogue systems is of wide interest, this paper is inappropriate for publication at ICML in my opinion. First, combining a text LLM with a speech encoder and a speech decoder has been done several times (Spectron, GLM-4Voice, Llama-Omni, etc.) and while this does not make the topic outdated, this paper does not present contributions in a way that convinces me that it does something fundamentally better than others: the model section is too vague (e.g. see below the relation between the NAR and AR modules) and the relation to previous work is not discussed enough to highlight the novelty of the framework. Second, the experiments are limited to ASR (as a proxy for speech encoder quality) and single-turn QA which are not sufficient to properly evaluate what claims to be a multi-turn spoken dialogue system. Finally, comparing Freeze-Omni to end-to-end models such as Moshi is a bit unfair: as Freeze-Omni uses a frozen LLM, it can be adapted easily, but the counterpart is that its text backbone acts as an information bottleneck that loses all non-linguistic information. Avoiding this limitation is the exact motivation behind end-to-end speech models! Overall, while quick adaptation of text LLMs to speech is of crucial interest both in research and in industry, the paper in its current state does not provide a convincing and replicable method and I strongly encourage the authors to submit a more detailed paper, with a more precise focus on the model and more extensive experiments in particular regarding multi-turn performance.

**Claims And Evidence:**

See "Summary".

**Essential References Not Discussed:**

N/A

**Experimental Designs Or Analyses:**

Overall, the experimental setup poses several problems:
1) The word error rates reported in Table 1 are much worse than sota in some cases. This may mislead the reader thinking that the proposed method is e.g. sota on Librispeech test-clean while sota on this dataset is below 1.5% WER.
2) Evaluating on question answering and emphasizing that Freeze-Omni gets a performance that is close to its LLM backbone is questionable: as the LLM is frozen, it is expected that answers will be identical to the textual topline as long as a) the speech encoder transcribes input audio properly b) the speech decoder produces speech that is intelligible enough to Paraformer for the transcribed answer to be verified against the ground-truth. This score here thus characterizes the performance of ASR and TTS (and probably to some extent the ability of the LLM backbone to correct ASR errors) which defeats the purpose of this metric, which is intended to measure the knowledge of end-to-end spoken LLMs.
3) The Q&A evaluation only evaluates single-turn interactions. As the authors perform training on multi-round sequences, they should provide metrics for multi-turn behaviour beyond the latency metrics reported in Table 4. Otherwise, it is not possible to favor their multi-turn setting rather than resetting the state of a single-turn dialogue model between every turns.
4) There are almost no ablation studies whatsoever. This does not allow identifying the key findings of authors through their experiments, and does not facilitate replication as it is unclear which components should be the most precisely reproduced.

**Methods And Evaluation Criteria:**

The writing of the method section could be considerably improved. In particular, I found the mechanism of the NAR and AR models to remain mysterious after carefully reading this section several times. On the other hand, some elementary descriptions such as Figure a are not useful to improve the understanding of the method.

**Other Comments Or Suggestions:**

N/A

**Other Strengths And Weaknesses:**

- The presentation of the paper is below ICML standards. As an example, the style of Figure 1 which is mostly empty space and with a large logo is unusual. Moreover, most acronyms are never defined. This is particularly problematic for not-so-standard acronyms such as NAR which afaik was introduced in Vall-E for "non-autoregressive". Moreover, the writing style can be improved, with many typos and inconsistent tense (e.g. Section 2.4).

**Questions For Authors:**

- Section 2.2.2. mentions that the second stage of training---which connects the pretrained speech encoder to the frozen LLM--- involves adding several trainable special tokens. What are those? How many of them, what separates them? How are they presented to the model and for which precise purpose?
- Section 2.3.1 explains that the speech generator combines a non-autoregressive (NAR) and an autoregressive (AR) model similarly to Vall-E. However, Vall-E first predicts the first codebook of the neural codec, while the NAR model generates the other levels. This is unlike Freeze-Omni which is claimed to first apply a NAR model, followed by an AR one. This would benefit from clarifications.
- Section 3.1.1 mentions that an LLM and a TTS systems are used to generate synthetic data. Which ones?

**Relation To Broader Scientific Literature:**

See "Summary"

**Theoretical Claims:**

None.

---

### Decision · Program_Chairs · 2025-05-01

**Decision:**

Accept (poster)

**Comment:**

This paper has got mixed reviews. The reviewer who gave a score 1 mainly complained about the presentation instead of technical aspects. The other weak score of 2 is mentions integrationg computer vision, which is out of scope for this paper. I am mainly agreeing with the reviewer who gave 4 and I think this paper must be presented in ICML.